# A New Sensory Approach Combined with a Text-Mining Tool to Create a Sensory Lexicon and Profile of Monovarietal Apple Juices

**DOI:** 10.3390/foods8120608

**Published:** 2019-11-22

**Authors:** Thais Mendes da Silva, Daniela Torello Marinoni, Cristiana Peano, Nicole Roberta Giuggioli

**Affiliations:** Department of Agricultural, Forest and Food Sciences, University of Torino, 10121 Torino, Italy; daniela.marinoni@unito.it (D.T.M.); cristiana.peano@unito.it (C.P.); nicole.giuggioli@unito.it (N.R.G.)

**Keywords:** sensory, fruit, projective mapping, quality, apple juice, text mining, sensory wheel

## Abstract

Single-cultivar juices may be a valuable way to introduce different versions of a product to the market and obtain price discrimination. To communicate a product’s value, complex characteristics incorporated by each cultivar must be identified. New sensory methods rely on the assessor’s ability to recall attributes; however, the use of objective vocabularies may improve the sensory profiling. This work aimed to profile monovarietal apple juices by using projective mapping (PM) combined with ultra-flash profiling (UFP) supported by a sensory wheel built with a text-mining tool. Samples were also analyzed for physicochemical parameters to provide more information to the assessment. The assessor coordinates from PM were used in multiple factor analysis with confidence ellipses to assess differences among samples. A goodness-of-fit test was applied to select the most meaningful descriptors generated through the UFP test by calculating the expected frequency of choosing a descriptor from the sensory wheel and comparing it with the observed values. The methodology provided a more accurate sensory profile compared to previous research on fresh apples and juices. Elstar, Jonagold, and Pinova were considered as sweet juices, and Gravensteiner was described as sour and astringent, with green-apple notes. Rubinette was described as having a strong taste and cloudy aspect.

## 1. Introduction

Increase of competitiveness in the internationalized juice market encourages European producers to develop strategies to create value [1] and differentiate their products [2]. Among all marketing strategies, product variation is key to segmenting consumer demand and obtaining price discrimination in the market [3]. It is well-known that different product versions may help to highlight a producer brand from those of others, and consumers may be less resistant to higher prices compared to the standard version of the same product [3]. In the case of the juice market, there is already a wide range of alternatives to purchase from, which leads to differences in loyalty levels in a brand [1].

In the apple juice market, use of local apple cultivars may be a valuable way to introduce product variation and to create a unique profile that is linked to a specific production context, thus creating quality attributes and promoting emotional meaning among consumers [2,4]. In monovarietal juices, typical features of particular cultivars, such as sweetness and aromatic notes, are usually marketed with the product [5]. Therefore, it is essential that complex sensory characteristics incorporated by each cultivar are identified and eventually communicated, because the degree of information available to consumers determines the value associated to the product [2].

There has been an intense development of sensory methodologies for mapping and profiling complex products that are known to be efficient alternatives to conventional methods (such as check-all-that apply and projective mapping), without requiring extensive training of assessors [6]. Many of these methods rely on the assessor’s ability to recall specific attributes to profile the product [7]. However, there is great evidence that the use of objective vocabularies in different sectors (industries or research departments) has led to a consistent improvement of product descriptions [8]. Objective vocabularies ensure better agreement among panelists since communication is made using the same reference [8]. In the case of apple juice, many authors have already developed a comprehensive lexicon for describing complex sensory attributes, although this information has not yet been exploited for further single-cultivar product characterization. Furthermore, no systematic way has been proposed to put together information from the literature to create a common lexicon for apple juice products. Among different automatic approaches, data mining is becoming a promising multidisciplinary field that involves informatics, statistics, and pattern recognition. One of its current aims is to analyze large amounts of data to find meaningful and useful information from the market, and to use it to identify new consumption trends. Among all methods involving data analysis, the text-mining process is an emerging technique that focuses on splitting and organizing a text into terms in order to analyze the most and least frequently used words and their associations. To the best of our knowledge, this tool has not yet been used at the level of sensory data, to improve a product’s characterization.

Thus, the aims of this work were twofold: to use emerging sensory methods to describe complex characteristics of six monovarietal apple juices and to build an apple sensory wheel to be used as a reference lexicon through a text-mining tool. Samples were also analyzed for physicochemical (total soluble content, titratable acidity, and dry matter) and colorimetric parameters with a chromameter, in order to improve quality information while relating it to the sensory attributes.

## 2. Materials and Methods

### 2.1. Physicochemical Parameters and Colorimetric Analysis

The six monovarietal apple juices from apples grown in south Tyrol and branded Kohl—Pinova, Gravensteiner, Rouge, Jonagold, Elstar, and Rubinette—were acquired from a Northern Italian local market. Total solid soluble (TSS), titratable acidity (TA), and colorimetric parameters were determined in triplicate from juice samples. TSS was measured with a digital refractometer (Atago, mod. PAL-1) and is expressed in percentage form, while TA was assessed by titration with 0.1 N NaOH to pH 8.1 and is expressed as g/100 g of malic acid. The ratio of TSS and TA (Ratio TSS/TA) was calculated, as well as the BrimA index, as proposed by others [9], for apple fruit. The BrimA index, which stands for “°Brix minus Acidity” is calculated using the following formula:
BrimA = Total soluble solids – (*k* × Titratable acidity)(1)
where *k* is a constant that may vary between fruit species/cultivars due to differing mixes of acids and sugars [10]. As suggested in previous research, the coefficient for apples is 10.

Dry matter (DM) was determined gravimetrically by heating all samples in a water bath at 100 °C for 4 h, in order to avoid boiling and water splashing, followed by drying the samples at 70 °C in the oven for 24 h. Samples were subsequently cooled in a desiccator, in order to be weighed at room temperature. The procedure was repeated until a constant weight was reached.

Results are expressed in percentage terms as follow:(2)[(C−A) /(B−A)] × 100
where *A* = weight of Petri dish; *B* = total weight of fresh sample + Petri dish; and *C* = total weight of dry sample + Petri dish.

The color of juice samples was measured with a handheld colorimeter (Konica Minolta, mod. CR-400). The parameters L*, a*, and b* were recorded with Konica Minolta software (SpectraMagic NX software). Derived from L*, a*, and b*, other color indexes previously tested were calculated in order to enhance sensitivity of color evaluation [11] as follows:(3)$$\mathrm{Chroma}\;(C*):\;\sqrt{a^{2}*+\;b^{2}*}$$
(4)$$\mathrm{Hue}\;\mathrm{angle}\;(h*):\;{tan}^{-1}\frac{b*}{a*}$$

The h* is a qualitative parameter of colorfulness since it reflects the visual color appearance with reference to a gray color with the same lightness. In the case of apple juice, it becomes redder when the hue decreases and yellowish when h* increases. The C* parameter is a quantitative parameter and is related with the color intensity or saturation since it expresses the degree of difference of a hue in comparison with a gray color with the same lightness [12].

### 2.2. The Text-Mining Tool and the Wheel Development

A list of descriptors was developed with a text-mining process, using the R package “tm” [13]. The “Material and Methods” and “Results and Discussion” sections of 17 peer-reviewed papers [2,5,8,14,15,16,17,18,19,20,21,22,23,24,25,26,27] related to sensory analysis of apple juice from 1998 to 2018 were selected from the Scopus database to be the “corpus” of the text-mining computation. Before texts could be assessed, a preprocessing step was necessary to reduce the amount of useless information of the corpus and transform the text into “tokens” (units of text such as words). This was required because full texts are too specific to perform meaningful computations [28]. In this work, this process was computed automatically to remove numbers, common words, capitalization, “stop words”, and punctuation. After the preprocessing step, the corpus was transformed to a document term matrix (dtm), a structured matrix with frequencies of terms where each row represents a document (uploaded articles) and each column represents a term. The dtm contained more than 1500 terms that appeared at least once, leading to a very high value of sparsity (>90%). This means that the dtm presented many empty cells, which is very common in a text-analysis process [13]. Therefore, terms that were not present in at least 10% of the documents were removed. This operation gave rise to a dtm composed of only 91 terms.

Filtering and normalization were also needed to further clean the corpus, select meaningful data, and to fuse words with identical meaning. It is important to note that those steps were mainly done manually, since text-mining tools do not take into consideration the meaning of the words, and efficiency of cleaning may eventually present some flaws, such as unremoved punctuation. Figure 1 demonstrates the network of the most frequent terms based on their co-occurrence in the selected articles prior to the manual filtering and the normalization step. It is evident that punctuation was not completely removed from the corpus.

Filtering and normalization are also considered critical because they can strongly influence the subsequent analysis, as they define the semantics of the texts [28]. Some authors may suggest to stem the text, a process called “lemmatization”, where a rule-based algorithm will convert inflected forms of words to their most essential meaning [28] so they can be recognized by computer (e.g., sour, sourness, and sourest to sour*). However, this process may sometimes be problematic, depending on the type of word (e.g., flowery to flow*) and, thus, in this work, it was not applied. In this work, expert knowledge from SATA S.r.l. and University of Turin was applied to filter and normalize the terms, avoiding loss of information and inclusion of bias. Furthermore, only descriptive data were selected, and affective data were not considered. Finally, this process also included the combination of occurring token pairs of tuples (e.g., green-apple), as suggested by other authors [29]. The normalization process provided a reduction from 91 to only 53 terms. Additional terms were introduced in order to complete the wheel information and properly organize the attributes.

### 2.3. Sensorial Analysis

Fifteen panelists—eight female and seven male—ranging from 22 to 35 years old, from SATA S.r.l. (Alessandria, Italy), with previous experience in sensory evaluation of fresh fruit were subjected to further theoretical training by discussing the definition of quality parameters concerning taste, texture, odor, and aromatic notes that were present in the developed sensory wheel. A quantity of 50 mL of each juice sample was codified with a 3-digit code and presented simultaneously, in random order, for each assessor. Assessors were asked to place each juice on a 60 × 40 cm sheet based on their similarities or differences. Individually, they were then asked to add from at least one to a maximum of five descriptors from the sensory wheel to each sample placed on the map, as a means to describe the sensory characteristics, following the projective mapping combined with the ultra-flash profile (PM-UFP) methodology [2]. For each sample, both X and Y coordinates were collected and compiled in a table, as described by Stolzenbach [2], along with the generated descriptors.

### 2.4. Statistical Analysis

Results from physicochemical and colorimetric data were analyzed with one-way analysis of variance (ANOVA), while results from sensorial analysis were analyzed with the multivariate multiple factor analysis for contingency tables (MFACT). The coordinates X and Y from each assessor of each product were treated as a group of two active variables to build the first two dimensions. Data were not scaled. Furthermore, 95% confidence ellipses were applied around the sample mean points, letting the bootstrap sequence iterate on the assessor’s partial (rotated) coordinates instead of the original assessor data, as suggested by other authors [30]. Using this approach, the confidence intervals do not include the assessor’s variability, since the objective is to compare the apple juice products.

The frequencies of terms to be used as supplementary variables in the MFA analysis following the UFP method were selected based on the common practice of choosing a descriptor that is cited no less than an arbitrary number of times by the panelists (classic approach) [2]; in this work, this was at least three times. In order to further emphasize the consensus and to determine when a particular descriptor was significantly selected to describe each single-cultivar juice sample, the goodness-of-fit test was used to assess how the observed frequency values for each sample were significantly different from the expected frequency values, where the expected frequency was considered to be the number of times a descriptor would be selected from random chance.

Considering the possibility of choosing “d” attributes that comprise the sensory wheel, and the possibility of choosing from one to five attributes for each sample, for each assessor, the expected frequency E was calculated as follows:(5)E = P × number of judges
where *P* is the probability that a descriptor is chosen from the sensory wheel by an assessor, calculated as the following:(6)$$P\;=\;\frac{C}{TC}$$
where *TC* is the number of possible combinations to select from 1 to n descriptors; and *C* is the number of possible combinations to select 1 specific descriptor from 1 to n descriptors. Therefore, *TC* and *C* were calculated as follows:(7)$$TC=\;\frac{d!}{1!\;\times \;\left(d-1\right)!}\;+\;\frac{d!}{2!\;\times \;\left(d-2\right)!}\;+\;\frac{d!}{3!\;\times \;\left(d-3\right)!}\;+\;\frac{d!}{4!\;\times \;\left(d-4\right)!}\;+\;\ldots \;+\;\frac{d!}{n!\;\times \;\left(d-n\right)!}$$
(8)$$C\;=\;\frac{\left(d-1\right)!}{1!\;\times \;\left(d-1-1\right)!}\;+\;\frac{\left(d-1\right)!}{2!\;\times \;\left(d-1-2\right)!}\;+\;\frac{\left(d-1\right)!}{3!\;\times \;\left(d-1-3\right)!}\;+\;\frac{\left(d-1\right)!}{4!\;\times \;\left(d-1-4\right)!}\;+\;\ldots \;+\;\frac{\left(d-1\right)!}{n!\;\times \;\left(d-1-n\right)!}$$

## 3. Results and Discussion

### 3.1. Physicochemical Parameters and Colorimetric Analysis

In Figure 2, it is possible to observe that the Rubinette juice was the sample with the highest level of TSS, while Gravensteiner had the lowest value. Pinova, Jonagold, Rouge, and Elstar presented intermediate values. The Rubinette sample also presented the highest value of TA, along with the Rouge juice, while there were no significant differences among the other juice samples, considering a confident interval of 95% level. Results of dry matter were consistent with TSS values for all samples (Appendix A).

It is well-known that differences in TSS alone do not have practical importance regarding consumer perception of fruit sweetness. Thus, it is very common to use the TSS/TA ratio to obtain better predictions of sweetness [10]. Apple fruit with TSS/TA ratios over 20 are considered to be sweet, while values under 20 are considered to be sour [9]. In this work, the values obtained for both TSS/TA ratio and BrimA reflect the sensory profile described in different articles [30,31,32]; that is, Jonagold, Pinova, and Elstar were ranked as the sweetest samples, while Gravensteiner, Rubinette, and Rouge were the least sweet. It is interesting to observe in Table 1 that values of the TSS/TA ratio and BrimA did not rank all samples in the same order. This might be due to BrimA’s characteristics of reducing the impact of acidity values in the final index value compared to TSS/TA ratio by only subtracting TA from TSS instead of dividing. This approach might be reasonable considering that small changes in acidity values lead to higher changes in taste perception compared to sweetness [33].

The TSS/TA ratio ranks the Jonagold sample as the sweetest, while BrimA indicates the Pinova sample. The same occurs for Gravensteiner and Rubinette samples, in which Gravensteiner would be considered less sweet than Rubinette, considering the TSS/TA ratio, while the opposite is true for the BrimA index. In this work, the BrimA index was introduced in order to properly evaluate the relationship between TSS and TA. This index takes into account differences in ratios of acids and sugars in different species by introducing a k-coefficient [33]. The main idea is that the human tongue perceives sugar and acids with differing sensitivities; thus, this index allows smaller amounts of acid to make the same numerical change to BrimA, as well as sugar [33,34,35]. Although different articles describe the sensory profile of the fresh-apple samples used in this work [36,37,38], no information is available regarding comparisons among Rubinette, Gravensteiner, Pinova, and Jonagold sweetness perceptions. Therefore, the use of the sensory analysis is the only possible way to verify if one of the two parameters is the best approximation of sweetness perception as a quality indicator.

Concerning the colorimetric assessment, in Figure 3, it is suggested that Jonagold has presented a distinct color concerning the degree of yellowness of the juice (demonstrated by higher values of b), and a more intense color compared to the other samples (demonstrated by higher values of C*).

By observing a* values (Figure 4), however, it is clear that Jonagold also presented a redder color, along with the Rouge sample. This probably affected the h* value which, by taking account of both a* and b* values, identifies Rubinette and Pinova as the yellowish-greenish samples, while Rouge and Jonagold were redder. Pinova and Rubinette also presented a brighter or lighter color compared with the same two samples, as suggested by L* values.

Assessing the complex index h*, along with the classic colorimetric values, enables the detection of different color patterns behind the term “lighter”, which were previously used indistinctly for both Jonagold and Pinova juices in other authors’ work [4]. As stated in the previous literature [35], the lower levels of anthocyanins of Pinova juice contribute to a less-red color and a lighter appearance of the sample. Thus, in this assessment, the level of anthocyanins of the Pinova sample could have contributed to its more yellowish color compared to the Jonagold juice, as indicated by the h* index measured in this work.

### 3.2. The Wheel Development

In this work, the goal of exploiting available data to build a sensory wheel by combining a text mining tool and expert knowledge was successfully achieved. Other than assisting in the text-mining process, the expert knowledge also aided reorganizing the selected terms in the wheel by checking the term’s correlations, generated as the covariance among the numeric vectors of the dtm (number of terms) divided by their standard deviation. By assessing the correlation among terms, it was possible to find how often two words appeared together relative to how often they appeared separately, independently of their meaning. This provided information for the normalization process that must take place in order to avoid redundancy generated by the use of words with identical or similar meanings. Other than grouping words based on their common root, related terms were fused when the correlation coefficient among them was over 0.50, including some of the volatiles involved in odor and aromatic notes, as well.

Table 2 summarizes how information was normalized and organized.

Once the normalization step was completed, the most cited terms were organized in descending order, as demonstrated in Figure 5.

Based on Figure 5, it is evident that apple taste, sweet, and sour are the most cited terms and the most important in describing apple juice. The text-mining tool also provided many attributes that are described as negative notes (such as medicinal, musty, smoky, and burnt), possibly from microbiological spoilage or technological issues during apple juice processing [24,39]. Therefore, considering the least-cited words is also important, in order to retain valuable information from the corpus. In this work, it is demonstrated how the text analysis can compile valuable data from the source; however, expert knowledge is still a fundamental resource to build an exhaustive system of information based on the available data. The sensory wheel is shown in Figure 6.

The resulted sensory wheel included the most- and least-cited terms in the selected bibliography, in order to cover all macro aspects (divided by different colors in the wheel) of apple juice quality. This process also included complex attributes, such as texture, which might be assessed through tactile sensations in the mouth or visually, based on the product’s appearance, as is demonstrated in Figure 6. Taste, which is the most cited macro attribute, includes qualitative (bitter, sour, and sweet) and quantitative (intensity and persistence) attributes. Usually, those descriptors are not present in conventional sensory methods, such as quantitative descriptive analysis (QDA) [6,7]. It was decided to retain this information in the assessment due to the relevance of these attributes in taste perception. To conclude, considering the use of assessors that have undergone theoretical training, astringent, metallic, and pungent were placed in the mouthfeel group, while apple taste was included under the odor and aromatic notes group, split into green- or ripe-apple attributes. However, it is important to note that this organization might be misleading in the context of a sensory evaluation at the consumer level, due to lack of knowledge concerning the meaning and the classification of sensorial terms.

### 3.3. Projective Mapping (PM) and Ultra-Flash Profile (UFP) Characterization

The PM method led to an acceptable differentiation of juice samples, as shown by the higher values of total explained variance (84.12%) in the first two dimensions Dim 1 and Dim 2 (Figure 7).

The higher percentage of explained variance (70.09%) of dimension 1 highlights the higher level of agreement among assessors in sample discrimination. Considering the rank obtained by the TSS/TA ratio and BrimA in Table 1, assessors distinguished two groups of samples, probably based on these parameters’ values over the first dimension. In fact, samples with higher values of TSS/TA ratio and BrimA are positioned on the right side of the individual plot, while samples with lower values are on the left side. Assessors were more prone to discriminate samples through the second dimension only in the case of juices with higher levels of TA.

In order to conduct the ultra-flash profile test, assessors were asked to choose from one to five samples of the sensory wheels. Assessors were also instructed to use the descriptors present on the outer ring of the wheel and, for the case of the group odor and aroma notes only, they were allowed to use the generic subcategories vegetable, fruity, toasted, and spicy. Therefore, following the calculations proposed in Equations (5)–(8), the possibility of choosing 46 different attributes from the sensory wheel led to the probability of choosing a particular descriptor by each assessor of 10% (Figure 8) and a rounded expected frequency, E, of two citations by descriptor by juice sample, considering all 15 assessors. As suggested in the literature, considering the low value of the expected frequency, the binomial test was used instead of the chi-squared test, to assess goodness-of-fit [40]. By assessing the *p*-values obtained using increasing values of frequency, the test suggests it is sufficient to have more than four citations for an attribute for a specific juice sample in order to consider it was not chosen by chance on the product description.

As is demonstrated in Figure 9, the amount of information retained with the classical approach is higher than the number of descriptors validated with the goodness of fit approach (Figure 10), and also includes negative descriptors of apple juice samples, especially in regard to the Rouge sample. However, there is a lack of coherence in the disposition of some descriptors. For example, the descriptor “bodied” is positioned in contrast with the descriptor “cloudy”, even though it is commonly accepted that cloudy apple juices are usually perceived as bodied [41], and “grassy” is close to samples described as “sweet” and far from those described as having a green-apple note, which is in contrast to the results of previous work concerning the aroma profile of apples using sensorial and chemical analyses [21,42]. Moreover, the distribution of descriptors in the MFA plot is not sufficiently spread to describe samples in a more specific way. The use of different scales between the active (coordinates) and the supplementary (frequencies) variables, and the low frequency of citations, led to a poor distribution of the descriptors compared to the samples in the multivariate space. Therefore, the selection of descriptors with the goodness-of-fit approach highlighted in Figure 10 is clearer and more coherent with the information available in the literature concerning fresh and juice samples of the cultivars used in this study.

The *p*-values obtained in the binomial tests suggest that Elstar, Jonagold, and Pinova were considered to be significantly sweet. Elstar presented the lowest *p*-value among all samples, indicating a higher number of citations on sweetness, suggesting that the aromatic profile of this cultivar [5] led to an increased perception of sweetness, despite intermediate values of this sample with regard to brix and both TSS/TA ratio and BrimA index. Jonagold presented a lower *p*-value compared to Pinova, which is more in agreement with the rank obtained with the TSS/TA ratio compared to the BrimA index, meaning that this parameter is a better approximation of sweetness perception. Therefore, this work confirms that the BrimA index in the case of apples may not improve the prediction of the sweetness perception over the TSS/TA ratio, as was suggested in previous work [10].

The sour attribute was significantly chosen only to describe the Gravensteiner sample, even though the TSS/TA ratio also indicated the samples Rubinette and Rouge as sour (values <20) samples. The values of the TSS/TA ratio of the three samples obtained in this work were similar, regardless if they were derived from higher concentrations of TSS and TA, as in the cases of Rubinette and Rouge, or lower concentrations of TSS and TA, as for Gravensteiner juice. This is a major drawback of the use of this parameter. Apparently, the higher amount of TSS present in Rouge and Rubinette samples was sufficient to contrast the sour taste induced by the higher concentration of TA, while the lower levels of TSS and TA of Gravensteiner led to taste discrimination in terms of sourness. If confirmed, at a consumer level, that this information is very important for the development of apple juice for specific segments of consumers that prefer a sour taste. Interestingly, regarding other taste attributes, the Rubinette sample was described as presenting a strong taste, which could be related to the higher proportions of TSS and TA values.

Concerning the apple juices’ appearance, the results obtained from the colorimetric assessment were partially comparable to the sensory assessment. Gravensteiner and Pinova were considered to present a lighter color, while, in regard to the colorimetric parameter L*, this is true also for Rubinette samples. Pinova and Elstar were considered to be the most yellowish, while the h* index also indicated the Rubinette juice. The higher level of cloudiness present in the Rubinette juice, demonstrated by the lower *p*-value with respect to the other samples, could have led to differences in the colorimetric measurements, since the colorimeter is only able to assess the color of samples through the reflectance modality [10]. The same may be applied to the Jonagold juice, which presented the second lowest value of L* and is considered to be the least cloudy sample, along with Elstar. Overall, some of the colorimetric values of samples classified as clarified were underestimated, while those of cloudy samples were overestimated. This has been shown in other work [5], as well. Thus, the introduction of different appearance attributes in the sensory assessment was important to assess the reliability of the colorimetric measurements done in this work. We suggest the use of a spectrophotometer when measuring liquids, since this also takes into account the opacity of products.

Considering the aromatic profile of the apple juices, the descriptions obtained in this work were reasonably similar to those described in the literature. Gravensteiner, which is known to present higher levels of α-farnesene, a compound that is related to herbaceous and green notes [38], was the only sample described with a green-apple note, and this is in agreement with the use of the sour descriptor for the taste profiling. Pinova, which is described as an apple having a fruity aroma [43], was profiled with a ripe-apple note, and Jonagold was described as a sample presenting a pear-like aroma, which is in agreement with the work of Jaros et al. [5,44], where the authors highlight the higher amounts of hexyl-acetate in this variety.

Finally, Rubinette was described as a pungent sample. Pungency in apple is associated with molecules that are both related to fruity (acetaldehyde and butanal) or green (hexenal) notes [45], but there is no available information considering the Rubinette sensory profile in order to confirm the obtained data. Astringency was found only for Gravensteiner and Rouge samples. It is not surprising that the Gravensteiner juice was described as astringent since this variety is characterized by higher values of polyphenols, which are known to be less present in the new varieties and may impart bitterness and astringency to food products [46].

## 4. Conclusions

In this work, the use of a sensory wheel, combined with new sensory profiling methods, projective mapping, and ultra-flash profiling, provided the characterization of six different apple juice samples, by taking into account almost all macro aspects present in previous sensorial works. The exploitation of available data through a text-mining tool process to construct the sensory wheel was successfully achieved, although this process must be complemented by expert knowledge, in order to filter and contextualize data into meaningful information and avoid bias and redundancy. Nonetheless, the goodness-of-fit test applied to the descriptors cited during the ultra-flash profiling provided a coherent selection of attributes for each apple juice sample, which resulted in a clearer description of each sample’s sensory profile and less contrast with previous work. This work also presented valuable information about how physicochemical characteristics in terms of acidity, brix, and their interaction may affect taste perception in apple juices. Secondly, we suggest colorimetric measurements still need to be improved in order to predict the perceived hue of juices’ color. The presented methodology is effective for future analysis and may be extended to other products where sensory information is available in the literature. Furthermore, we highlight that theoretical training concerning the selected attributes to be incorporated in the sensory wheel is fundamental in order to promote proper use, especially in the case of complex attributes, such as texture.

## Figures and Tables

**Figure 1 foods-08-00608-f001:**
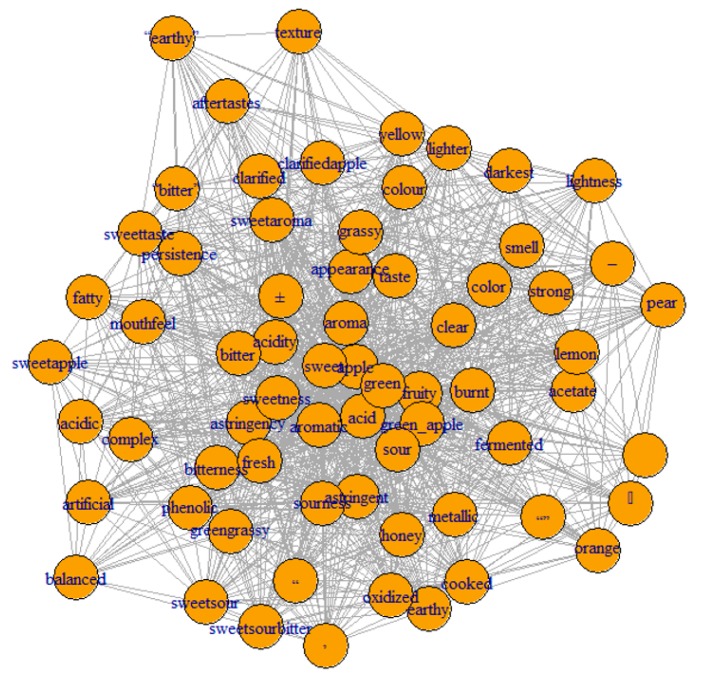
Network of most frequent terms based on their co-occurrence in the selected articles before the manual filtering and the normalization step.

**Figure 2 foods-08-00608-f002:**
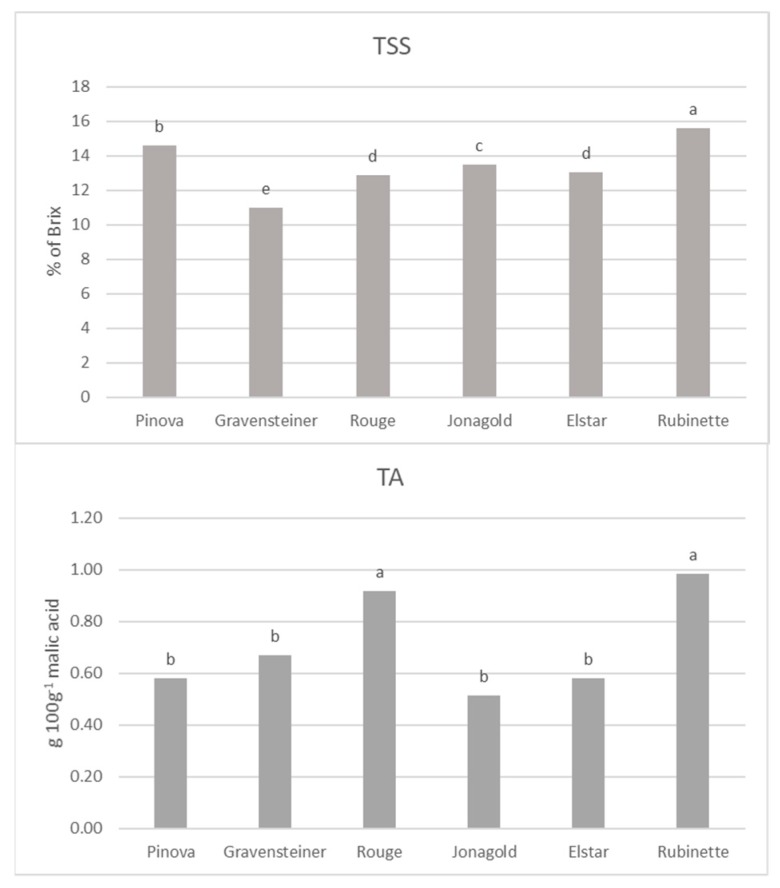
Total soluble solids (TSS) (**2a**) and Titratable acidity (TA) (**2b**) values of apple juice samples. Different lower-case letters (a–d) show significant differences among treatments (*p*-value ≤ 0.05).

**Figure 3 foods-08-00608-f003:**
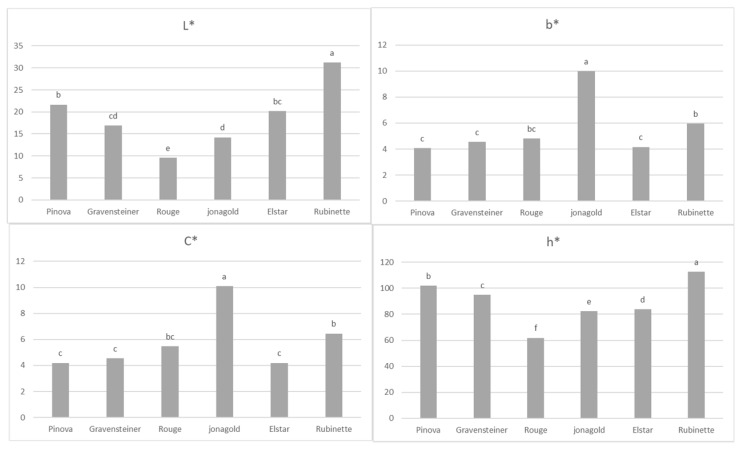
L*, b*, Chroma (C*), and hue angle (h*) As stated in the previous literature [35], the lower levels of anthocyanins of Pinova juice contribute to a less-red color and a lighter appearance of the sample. Thus, in this assessment, the level of anthocyanins of the Pinova sample could have contributed to its more yellowish color compared to the Jonagold juice, as indicated by the h* index measured in this work.

**Figure 4 foods-08-00608-f004:**
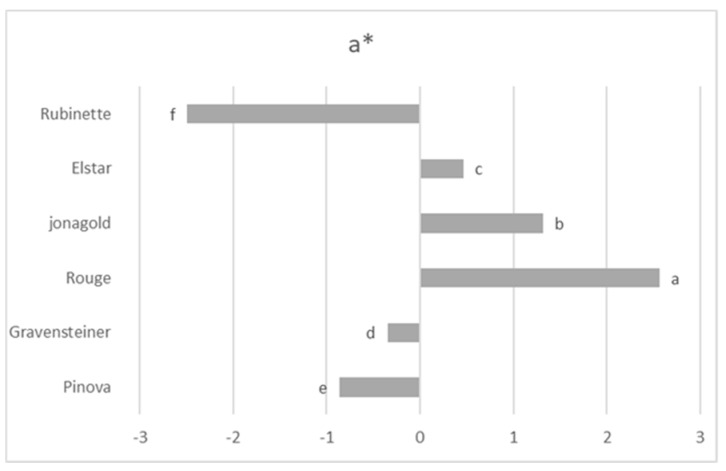
The a* values of apple juice samples. Different lower-case letters (a–f) show significant differences among treatments (*p* ≤ 0.05).

**Figure 5 foods-08-00608-f005:**
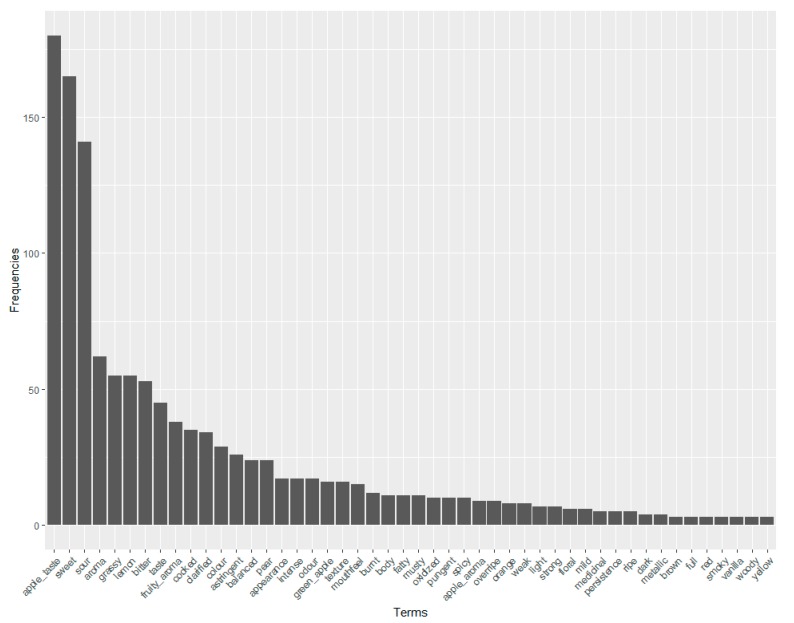
Frequency of the selected terms obtained from the text mining and normalization process.

**Figure 6 foods-08-00608-f006:**
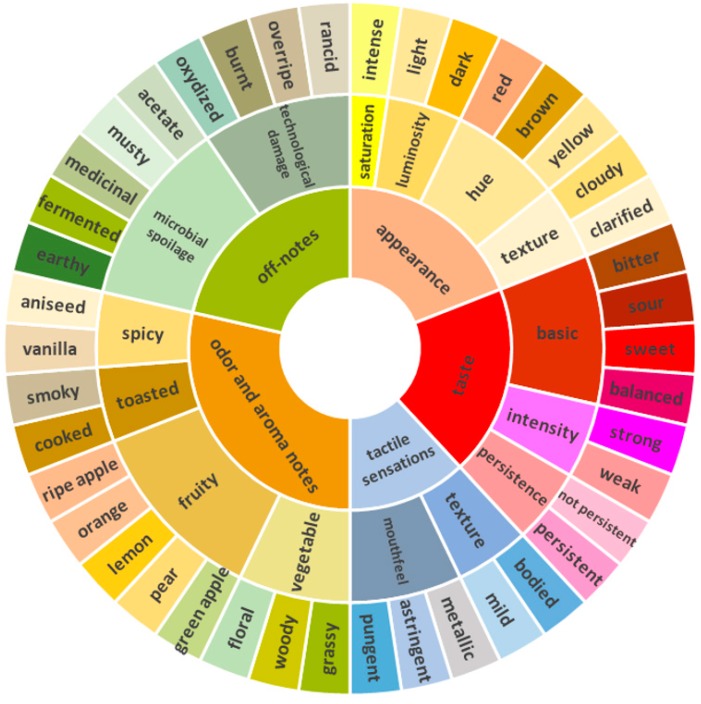
Sensory wheel developed with sensory attributes selected from the text-mining process.

**Figure 7 foods-08-00608-f007:**
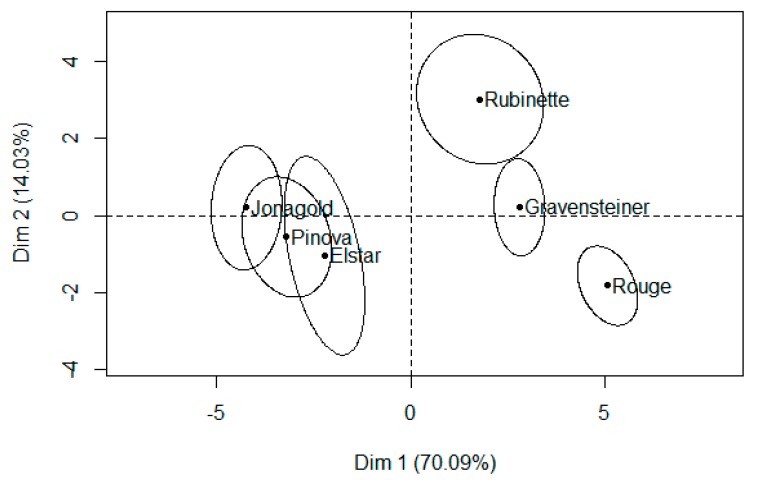
Dimension 1 (Dim 1) and Dimension 2 (Dim 2) of the multiple factor analysis individual plot of apple juice samples and confidence ellipses.

**Figure 8 foods-08-00608-f008:**
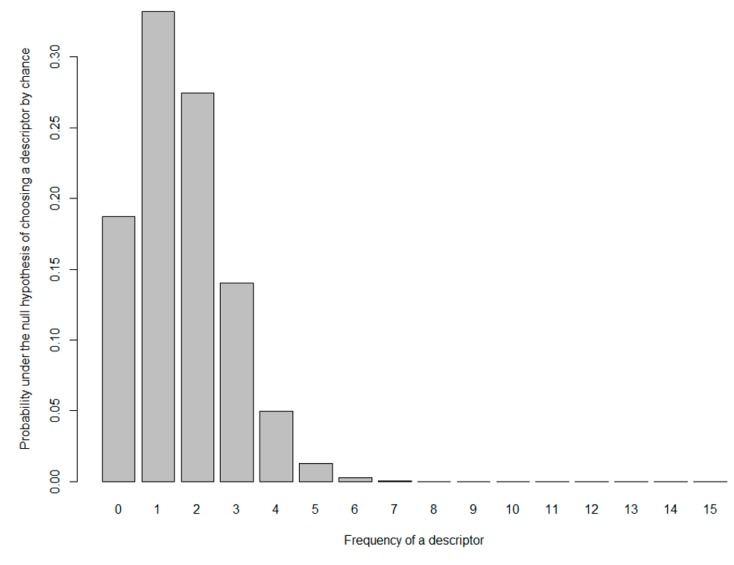
The expected probability distribution for the binomial with 15 trials (number of assessors) considering the probability of 10% of choosing a descriptor from the developed sensory wheel.

**Figure 9 foods-08-00608-f009:**
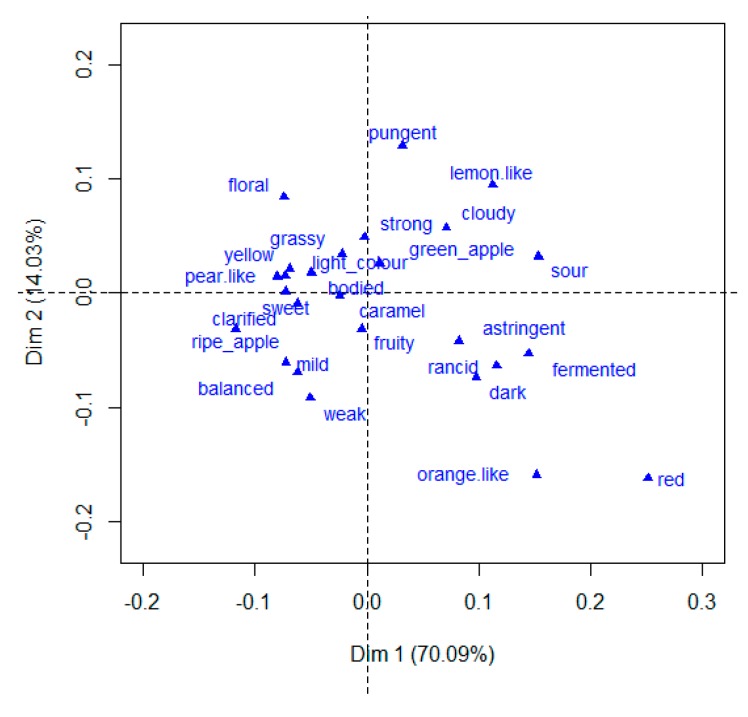
Multiple factor analysis plot of descriptors selected through the classic approach.

**Figure 10 foods-08-00608-f010:**
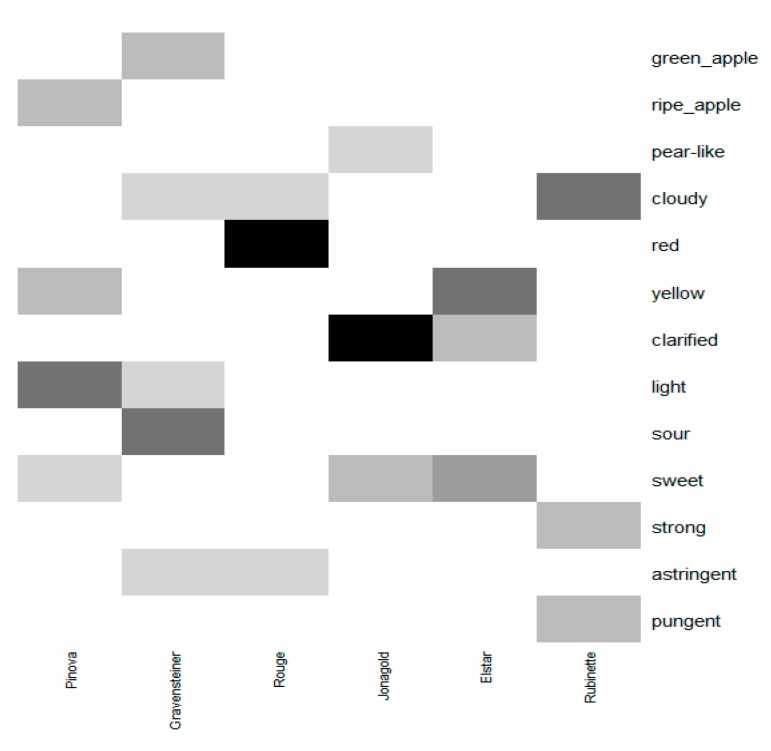
Heatmap of *p*-values < 0.05 obtained for each product and each descriptor from the binomial test. Darker colors indicate lower *p*-values (<0.000001), and lighter gray colors indicate higher *p*-values (<0.01).

**Table 1 foods-08-00608-t001:** Ratio total soluble solids/titratable acidity and BrimA values of apple juice samples and their relative rank, in descending order.

	Ratio TSS/TA	Rank	BrimA	Rank
Pinova	25.1	2	8.8	1
Gravensteiner	16.4	4	4.3	5
Rouge	14.1	6	3.7	6
Jonagold	26.2	1	8.3	2
Elstar	22.5	3	7.2	3
Rubinette	15.9	5	5.8	4

**Table 2 foods-08-00608-t002:** Selected terms that underwent the normalization process with similar and related words.

Normalized Terms			
Term	Similar Terms	Related Terms *	Frequency
sweet	sweeter	/**	165 [2,5,14,15,16,17,18,20,21,22,23,24,25,26,27]
	sweetness	/	
sour	acid	/	141 [2,5,14,15,16,18,20,21,22,23,24,26,27]
	acidic	/	
	acidity	/	
	sourness	/	
aroma	aromatic	/	62 [2,5,14,15,16,20,22,24,25,26,27]
grassy	/	green	55 [2,14,15,17,18,21,22,26,27]
	/	phenolic	
	/	acetaldehyde	
lemon		hexanal green	55 [2,14,15,18,21,22,26,27]
bitter	bitterness	artificial	53 [5,14,16,18,20,21,22,23,24,27]
taste	tastes		45 [14,15,16,20,22,24,26]
cooked	/	candy	35 [14,17,18,26,27]
	/	caramel	
	/	honey	
clarified	clear	/	34 [5,14,20,22,24,27]
color	color	/	29 [5,14,16,20,22,23]
astringent	astringency	/	26 [14,15,18,20,21,22,27]
pear	pear-like	/	24 [16,18,20,22,23]
balanced	/	complex	24 [2,16,20,22,26,27]
	/	sweet.sour sour.sweet	
odor	odor	/	17 [15,17,24,26,27]
	smell	/	
light	lightness	/	7 [5,22,24,26]
floral	/	flowery	6 [18,27]
dark	darkest	/	4 [16,22]

* The related terms were selected based on a correlation coefficient > 0.50. ** “/” indicates absence of related terms.

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
