# Peer review of "A New Sensory Approach Combined with a Text-Mining Tool to Create a Sensory Lexicon and Profile of Monovarietal Apple Juices"

_foods, 2019, doi:10.3390/foods8120608_

Round 1
Reviewer 1 Report
The presented study is interesting, methodology well set and the presented results well described.
Some minor corrections
line 70: I suggest to change the equation as follows % DM= [(C-A)/(B-A)]*100
Line 98: please change (Feinerer, 2018) to [12]
line 174: change [18-19-20-21-22-23) to [18-23]
Fig. 3 add * next to L and b in the graphics
line 365_ change 5- to 5.
Author Response
Point 1: I suggest to change the equation as follows % DM= [(C-A)/(B-A)]*100.
Response 1: The equation was changed as required (line 80)
Point 2: please change (Feinerer, 2018) to [12]
Response 2: The citation was changed as required (line 95)
Point 3: change [18-19-20-21-22-23) to [18-23]
Response 3: The citation was changed as required (line 185)
Point 4: add * next to L and b in the graphics
Response 4: The figure was changed as required
Point 5: line 365_ change 5- to 5
Response 5: We have eliminated the number, as it is displayed in the journal's template.
Reviewer 2 Report
See attached.

Author Response
Response to Reviewer 2 Comments
Point 1: The results and discussion chapter needs revision and streamlining to make it clear and direct to the point. The organization and flow of discussion need to be revisited because as is, it is somewhat confusing.
Response 1: We have added more information where the text was not sufficiently clear as pointed by the reviewer in the next sections, and the manuscript went to English editing service. Please check again the chapter “Results and Discussion”.
Point 2: The discussion must highlight the samples with respect to the development of the sensory wheel and not the process of wheel development. Discussing the wheel development is somewhat boring to read. Should be discussed in context with the apple juice samples
Response 2: The development of the sensory wheel was done in this work as a result of the text mining process, which required an important effort of interpretation and decisions, including the illustration of the text mining outputs (tables and figures). Thus, we think it is more appropriate to describe the text mining process in Materials and Methods and explicit its results in the Discussion in order to highlight the new approach that we are proposing and to not extend excessively the Materials and Methods chapter with interpretation notes rather than the objective description of what we have done. The development of the sensory wheel was a previous step that did not take into account the sample´s nature and we think samples were extensively described in the next chapter of this work with the descriptors coming from the sensory wheel. We could have provided more information about the impact of using the wheel to describe samples if we had compared this approach with another one (without the wheel) which unfortunately was not possible to do in this work. Please note that our goal is to propose a new methodology and not only describe the samples.
Point 3: What makes the sensory wheel unique and special relative to the other sensory wheels developed by others
Response 3: To our knowledge, most of sensory wheels proposed in the food sector consider usually only aromatic notes and include a huge number of descriptors that might not be used by assessors. In our work, we wanted to exploit the available information of sensory descriptors for apple juices organising it on a tool that was attractive to panellists (compared to questionnaires) and retain only important descriptors, which were selected with the text mining tool and expert knowledge. We are not only proposing a wheel as a final product, but also the methodology to create it.
Point 4: The conclusion is a little bit weak. It has to be revised and highlight if the sensory wheel is effective or not
Response 4: The conclusion was changed as required Please check lines 386-401.
Point 5: There isn´t any flow in the description of the methodology used, so it is difficult to follow the procedure. Revise it
Response 5: The abstract was changed as required. See lines 13-18.
Point 6: Is this the methodology referred to in lines 20-21?
Response 6: Yes, we have altered the description as requested in the previous point.
Point 7: Lines 62, 74 and 81- colorimetric analysis comments (“of the flesh, of the skin? Colour of what?”)
Response 7: We have analysed only the juice samples, as described in line 68, and not the fresh fruit. We have added this information in lines 68 and 86.
Point 8: Line 65- describe what brimA is
Response 8: We have decided to describe BrimA in the discussion section since it is important to recall its meaning when assessing this index as a potential indicator of sweetness.
Point 9: Lines 66, 67 and 68- dry matter analysis comments
Response 9: We have added more information about the adopted method as requested. The procedure was done in respect to the Italian legislation for fruit juices. Please check lines 65-68
Point 10: Line 87. Reference the articles?
Response 10: We agree we have to provide this information. We have added the list of articles as an appendix of this work in order to not exceed the number of citations of the article. Pleas check lines 406-461.
Point 11: Line 89. Corpora?
Response 11: The English editing has corrected to corpus. Please check line 98.
Point 12: Lines 122 to 130. sensory method comments
Response 12: We have added more information about the sensory evaluation, as requested. The panel was already trained on fresh fruit but the assessment described in the article is only on apple juice. We wanted to highlight that our panel can be considered a semi trained panel, with no direct practical experience on the product that is being assessed. Please check lines 132-146.
Point 13: Line 167. How were you able to determine this? There are no error bars in the graphics
Response 13: Significant differences were determined with Anova analysis. We decided to not introduce the error bars since the standard deviations of most of our analyses were very low, including the TA one. Furthermore, this result is expected since juice samples are usually much more standardized than fresh fruit.
Point 14: Line 175. Why is this result?
Response 14: We have extensively explained differences among both indicators in lines 197 to 209.
Point 15: Comments on Figure 2: Graphic TSS and TA
Response 15: We have corrected figures as requested.
Point 16: lines 194 and 195. So, what´s the significance?
Response 16: We tried to also verify which physicochemical parameters could be a better-quality indicator of samples. We have added this information in the article in lines 207 and 209.
Point 17: colorimetric graphics?
Response 17: We would like to avoid the error bars since the standard deviation is very low. We may not add units to colorimetric parameters because they don’t present one.
Point 18: lines 202 and 203. Considerations about anthocyanins and polyphenol content
Response 18: We have considered the presence of those compounds in lines 229 to 231. We think it is not useful to add information about pigment loss during process since all juices are coming from the same industry, which uses the same process in all samples.
Point 19: “I don’t see the point of extensively discuss Lab values unless a standard is being used”.
Response 19: Unfortunately, no standards are being used for apple juices due to differences that might exist among different industrial processes. However, in our case, all samples were treated with the same process. Considering there is also a lack of information about colorimetric measurements of juices in literature, we think our work is providing valuable information about this topic.
Point 20: Line 216. The wheel development
Response 20: We think this is a very important part of our work, because it introduces novelty in the way our sector may exploit available information present in literature (or internet). There is a huge trend in the consumer science about this approach to find trends in the food sector, while we are proposing to use it to solve a different type of problem, which is related to the sensory analysis and its new methods. We are also explaining this process in detail in order to other people reproduce it. Thus, it is not only about profiling the juices. We would like to keep it.
Point 21: table 2. references
Response 21: references are provided in the appendix A.
Point 22: line 240
Response 22: we would like to keep this information in the article. Most of text mining approaches are done in order to analyse and keep terms that are highly cited in bibliography or internet. Few of them highlights the importance to assess also less frequent terms. This is the reason why we claim that the process must be supervised by expert knowledge, in lines 256 and 257.
Point 23: figure 6. description of sensory wheel
Response 23: we have added the requested information in lines 264 to 277.
Point 24: line 273. MFA plot
Response 24: we have added the requested information in lines 282 to 283.
Point 25: line 299 to 308. Discussion of BrimA and ratio TSS/TA
Response 25: we think the discussion should remain in this point because only at this point we can make considerations about the relationship among those indexes and the sweetness perception obtained in the work.
Point 26: line 301. specific results
Response 26: we don’t understand what the Reviewer points as a less specific result. We think the results coming from the sensorial analysis are specific. We are claiming at that paragraph that sensorial results didn’t completely relate with brimA and ratio TSS/TA values. The use of word ‘probably’ at line 334 is necessary because we did not include aroma analysis in this assessment, thus, we are supposing that the higher Elstar sweetness perception is due to its superior aromatic profile found in literature.
Reviewer 3 Report
line 18: THE most meaningful
line 20: a more accurate sensory profile compared to...
line 30: It is
line 32: in the case of the juice
line 36: clarify specific area
line 37: what does the asterisk indicate?
line 39: can remove incorporated by and just add "in"
line 44: rely
line 47: refs to back this up
line 53: the aim of this work was twofold
line 55: a bit more info should be given on text mining
line 67: not sure you need "a previous"
line 68: do you have a ref for this method?
line 71-73: you can add these together in the same line. "where a=xx, b=cc etc"
line 74: handheld colorimeter?
line 81: in the case of
line 87: Discussion
line 88: remove relevant on and add related to
line 89: could be assessed
line 99: in at least
line 104: say what figure one shows
Fig 1: remove previous, add before
Fig 1: I think it would be better to show the final one with 53, not 91
line 113: in this work,
line 117: from 91
line 117: add name before (15)
line 123/126: keep consistent with the numbers: 3, 5, one, five
line 127: on the map
line 141: define 141
line 144: 3 or three, make the whole manuscript consistent
line 145: was not is
line 146: were not are
line147: was not is
line 149: "d" attributes that compose
line 165: add Rouge to the list
line 169: it is, TSS alone do
line175: least sweet
fig 2: define tss and ta, remove asterisk and bring line 179 up, letters a-d
table 1 and 2: remove vertical rules, use appropriate Foods template
line 196: remove has
fig 3 and 4: remove asterisk and bring 201/209 up
line 213: contribute
line 219: also reorganized
line 222: possible to find
line 223: provides
line 226: fused, was over 0.50
line 230: 0.50
line 235: in fig.5
line237: tool allowed also to
line 243: knowledge is still, exhaustive
line 244: wheel is shown in fig 6.
fig 6: what do the colours indicate? Some terms on the wheel are capital, correct
line 254: it is
line 255 one. Considering
line 259: 1-5?
line 277 and the rest: decide if all the attributes are in "" or not
line 297/298: write p values more clearly, < 0.00..
line 300: to be significantly sweet
line 310: remove has
line 311: 3?
line 321: concerns
322: other work or others' work, attributes other than appearance
338: Capital F needed?
line 345: ref after hexanal
Author Response
Response to reviewer 3.
*Please note the manuscript has undergone english editing process, thus, all comments regarding english corrections won't be addressed to the lines where corrections were made.
line 18: THE most meaningful
#The line was corrected as requested.
line 20: a more accurate sensory profile compared to...
#The line was corrected as requested.
line 30: It is
#The line was corrected as requested.
line 32: in the case of the juice
#The line was corrected as requested.
line 36: clarify specific area
#The line was corrected as requested. Please check line 36.
line 37: what does the asterisk indicate?
#The line was corrected as requested. It was a mistake.
line 39: can remove incorporated by and just add "in"
#We prefer to keep incorporated in order to highlight the cultivar input that is intensionally introduced in the juice.
line 44: rely
#The line was corrected as requested.
line 47: refs to back this up
#The line was corrected as requested.
line 53: the aim of this work was twofold
#The line was corrected as requested.
line 55: a bit more info should be given on text mining
#We have added more information about it in the introduction section, lines 53-60. We also point that the method is extensively described in the nex chapters of the article.
line 67: not sure you need "a previous"
#The line was corrected as requested.
line 68: do you have a ref for this method?
#The method is an adaptation of the italian legislation, but since it was modified we prefer to describe it without citing.
line 71-73: you can add these together in the same line. "where a=xx, b=cc etc"
#The line was corrected as requested.
line 74: handheld colorimeter?
#Yes, we have added this information.
line 81: in the case of
#The line was corrected as requested.
line 87: Discussion
#The line was corrected as requested.
line 88: remove relevant on and add related to
#The line was corrected as requested.
line 89: could be assessed
#The line was corrected as requested.
line 99: in at least
#The line was corrected as requested but the english editing removed the correction.
line 104: say what figure one shows
#We have added this information in lines 113-115.
Fig 1: remove previous, add before
#The line was corrected as requested.
Fig 1: I think it would be better to show the final one with 53, not 91
#The network can only be done using the terms that are extracted in the document term matrix provided by the text mining tool, otherwise its relationships are not calculated based on the co-occurrence found in the documents that are part of the corpus. The final one with 53 terms is an manually organized table where track of original documents has been lost.
line 113: in this work,
#The line was corrected as requested.
line 117: from 91
#The line was corrected as requested.
line 117: add name before (15)
#The line was corrected as requested.
line 123/126: keep consistent with the numbers: 3, 5, one, five
#The line was corrected as requested.
line 127: on the map
#The line was corrected as requested.
line 141: define 141
#The line was corrected as requested.
line 144: 3 or three, make the whole manuscript consistent
#The line was corrected as requested.
line 145: was not is
#The line was corrected as requested.
line 146: were not are
#The line was corrected as requested.
line147: was not is
#The line was corrected as requested.
line 149: "d" attributes that compose
#The line was corrected as requested.
line 165: add Rouge to the list
#The line was corrected as requested.
line 169: it is, TSS alone do
#The line was corrected as requested.
line175: least sweet
#The line was corrected as requested.
fig 2: define tss and ta, remove asterisk and bring line 179 up, letters a-d
#The line was corrected as requested.
table 1 and 2: remove vertical rules, use appropriate Foods template
#The tables were corrected as requested.
line 196: remove has
#The line was corrected as requested.
fig 3 and 4: remove asterisk and bring 201/209 up
#The figures were corrected as requested.
line 213: contribute
#The line was corrected as requested.
line 219: also reorganized
#The line was corrected as requested.
line 222: possible to find
#The line was corrected as requested.
line 223: provides
#The line was corrected as requested.
line 226: fused, was over 0.50
#The line was corrected as requested.
line 230: 0.50
#The line was corrected as requested.
line 235: in fig.5
#The line was corrected as requested.
line237: tool allowed also to
#The line was corrected as requested.
line 243: knowledge is still, exhaustive
#The line was corrected as requested.
line 244: wheel is shown in fig 6.
#The Wheel was corrected as requested.
fig 6: what do the colours indicate? Some terms on the wheel are capital, correct
#We have added more information in the article. Line 265
line 254: it is
#The line was corrected as requested.
line 255 one. Considering
#The line was corrected as requested.
line 259: 1-5?
#The line was corrected as requested.
line 277 and the rest: decide if all the attributes are in "" or not
#The line was corrected as requested.
line 297/298: write p values more clearly, < 0.00..
#The line was corrected as requested.
line 300: to be significantly sweet
#The line was corrected as requested.
line 310: remove has
#The line was corrected as requested.
line 311: 3?
#we do not understand what the reviwer points
line 321: concerns
#The line was corrected as requested.
322: other work or others' work, attributes other than appearance
#The line was corrected as requested. The second correction was made keeping the original meaning.
338: Capital F needed?
#No, the line was corrected to "f".
line 345: ref after hexanal
#We have added the requested reference. Line 378.
Round 2
Reviewer 2 Report
See comments and revisions on the attached file.

Author Response
Dear Editor,
all recomended corrections were provided in the manuscript.
Please check lines:
75-79 for BrimA definition and T of water bath
100 and Reference sections for the added references concerning the sensory wheel
185 for the information concernin the confidence interval
195-198 for the requested consideration of why ranks TSS/TA ratio and BrimA are different
Figure 2 and table 2 were also modified with the requested corrections and references
244 for the anthocianin statement
306 for the correction of MFA plot interpretation
Conclusions were also improved with your corrections
Thank you and best regards,
Thais Mendes da Silva